# Pregnancy Outcomes in Women after the Fontan Procedure

**DOI:** 10.3390/jcm12030783

**Published:** 2023-01-18

**Authors:** Agnieszka Bartczak-Rutkowska, Lidia Tomkiewicz-Pająk, Katarzyna Kawka-Paciorkowska, Natalia Bajorek, Aleksandra Ciepłucha, Mariola Ropacka-Lesiak, Olga Trojnarska

**Affiliations:** 11st Department of Cardiology, Poznan University of Medical Sciences, 61-848 Poznan, Poland; 2Institute of Cardiology, Jagiellonian University Medical College, 31-202 Krakow, Poland; 3Department of Perinatology and Gynecology, Poznan University of Medical Sciences, 60-535 Poznan, Poland; 4Department of Medical Education, Centre for Innovative Medical Education, Jagiellonian University Medical College, 30-688 Krakow, Poland

**Keywords:** congenital heart disease, Fontan procedure, single ventricle, miscarriage, pregnancy, prematurity

## Abstract

Women with single ventricle physiology after the Fontan procedure, despite numerous possible complications, can reach adulthood and give birth. Pregnancy poses a hemodynamic burden for distorted physiology of Fontan circulation, but according to the literature, it is usually well tolerated unless the patient is a “failing” Fontan. Our study aimed to assess maternal and fetal outcomes in patients after the Fontan procedure followed up in two tertiary Polish medical centers. We retrospectively evaluated all pregnancies in women after the Fontan procedure who were followed up between 1995–2022. During the study period, 15 women after the Fontan procedure had 26 pregnancies. Among 26 pregnancies, eleven ended with miscarriages, and 15 pregnancies resulted in 16 live births. Fetal complications were observed in 9 (56.3%) live births, with prematurity being the most common complication (*n* = 7, 43.8%). We recorded 3 (18.8%) neonatal deaths. Obstetrical complications were present in 6 (40%) out of 15 completed pregnancies—two (13.3%) cases of abruptio placentae, two (13.3%) pregnancies with premature rupture of membranes, and two (13.3%) patients with antepartum hemorrhage. There was neither maternal death nor heart failure decompensation during pregnancy. In two (13.3%) women, atrial arrhythmia developed. One (6.7%) patient in the second trimester developed ventricular arrhythmia. None of the patients suffered from systemic thromboembolism during pregnancy. Pregnancy in women after the Fontan procedure is well tolerated. However, it is burdened by a high risk of miscarriage and multiple obstetrical complications. These women require specialized care provided by both experienced cardiologists and obstetricians.

## 1. Introduction

The Fontan procedure is one of the most inventive cardiosurgical concepts. It enables patients with complex congenital heart defects unsuitable for biventricular repair to reach adulthood. Fontan palliation through the complete separation of pulmonary and systemic circulation alleviates cyanosis but creates a challenge for the cardiovascular system [1]. Lack of the subpulmonary ventricle leads to nonpulsatile pulmonary blood flow that depends on low pulmonary vascular resistance and high systemic venous pressures. This solution tremendously improves the survival of these patients but also leads to numerous complications. Single ventricular dysfunction, refractory arrhythmia, Fontan-associated liver dysfunction, thromboembolic events, plastic bronchitis, protein-losing enteropathy, and cognitive disorders are examples of Fontan circulation failure. [2,3]. However, women who reach childbearing age desire to have children. Pregnancy is a hemodynamic burden to the normal cardiovascular system. At a six-week pregnancy, both heart rate and stroke volume start to increase, leading to a 30–50% rise in cardiac output [4]. As a result of progesterone, nitric oxide, and prostaglandin actions, there is a decrease in systemic and pulmonary vascular resistance. Volume loading results in hemodilution and physiologic anemia. Additionally, there is a rise in prothrombotic factors during pregnancy. All these changes challenge the distorted univentricular heart after the Fontan procedure [5]. Moreover, high systemic venous pressures and limited reserve to increase cardiac output may impact placental perfusion [6].

As per European Society of Cardiology (ESC) guidelines for the management of cardiovascular diseases during pregnancy, uncomplicated Fontan patients belong to modified World Health Organization (WHO) class III, which means a significantly increased risk of maternal mortality or severe morbidity [7]. However, patients called failing Fontans who demonstrate the above-mentioned complications of Fontan circulation are classified as a modified WHO class IV and should be counseled against pregnancy.

Our study aimed to assess maternal and fetal outcomes in patients after the Fontan procedure followed up in two tertiary Polish medical centers.

## 2. Materials and Methods

In this retrospective two-center (First Department of Cardiology, University of Medical Sciences, Poznan, and Institute of Cardiology, Jagiellonian University Medical College, Krakow) observational study, we evaluated all pregnancies of women after the Fontan procedure who were followed up between 1995–2022. We did not include ongoing pregnancies. Information from medical records included demographic data, initial heart anatomy, prior surgical procedures, cardiac complications before pregnancy, the latest echocardiography examination, age at first pregnancy, week, and mode of delivery. We also analyzed the number of patients’ visits to the outpatient clinic during pregnancy.

Echocardiography (Vivid 9, GE Healthcare, Wauwatosa, WI, USA) was performed by two experienced echocardiographers (ABR and LTP) according to a predetermined study protocol. The echocardiograms included 2D and Doppler imaging to identify the morphology of the single ventricle—dominant left, right, or mixed ventricle and the presence of thrombus. Based on biplane modified Simpson’s rule, the ejection fraction (EF) of a single ventricle was calculated. Impaired ventricular function was considered when EF was <50% [8]. Color-mode Doppler determined atrioventricular valve (AVV) regurgitation in a 4-chamber view.

Fetal outcomes were recorded and included: termination of pregnancy before 20 weeks of gestation (WG), miscarriage as spontaneous fetal loss before 24 WG, stillbirth as fetal death after 24 WG [9], premature delivery (delivery before 37 WG), small size for gestational age (SGA) (birth weight below 10th percentile), neonatal death (death within the first month after birth), and diagnosis of congenital malformations.

Maternal complications included: maternal death, heart failure, systemic thromboembolic complication, new cyanosis (SO_2_—oxygen saturation drop below 90% at rest), protein-losing enteropathy, and atrial or ventricular arrhythmia. Obstetrical complications were classified as preeclampsia, abruptio placentae, premature rupture of membranes before 37 WG, and antepartum and postpartum hemorrhage. Postpartum hemorrhage was defined as a loss of >1000 mL of blood after cesarean section until 24 h postpartum [10].

We also calculated the risk scores for adverse cardiac complications during pregnancy according to ZAHARA and CARPREG II scores [11,12].

As approved by our institutional Ethics Committee, the study protocol conformed to the ethical guidelines set forth by the 1975 Declaration of Helsinki.

## 3. Statistics

Data were expressed as means with SD or medians with range (according to normal distribution) for continuous variables and percentages for categorical variables for descriptive analysis. Analysis was performed using PQStat v.1.8.2. (PQStat Software, Poznan/Plewiska, Poland).

## 4. Results

### 4.1. Baseline Characteristics

During the study period, 15 women after the Fontan procedure had 26 pregnancies. Among them, three women had two pregnancies, two patients had three pregnancies, and one woman had five pregnancies. The mean age (SD) at first pregnancy was 25.4 (3.5) years. Underlying congenital heart defects and baseline maternal characteristics are presented in Table 1.

All palliations, except one—atrio-pulmonary connection—were total cavo-pulmonary connections (TCPC) performed at the mean age (SD) of 7.1 (4.4) years. Fenestrated TCPC was performed in six (40%) patients. One fenestration was closed with an Amplatzer device six years after Fontan’s completion. Hypoxemia was observed before pregnancy in one (6.7%) patient with patent fenestration (SO_2_—87%). All pregnancies were delivered by cesarean section.

### 4.2. Fetal Outcomes

Among 26 pregnancies, 11 ended in miscarriage at the median gestational age of 9.3 (5–18) WG, and the mean age (SD) of patients was 24.1 (3.1) years. Fifteen pregnancies resulted in sixteen live births at the median gestational age of 35.5 (26–40) WG and the mean age (SD) of patients of 26.3 (3.8) years. The mean birth weight (SD) was 2519 (758) g. Fetal complications were observed in 9 (56.3%) live births, with prematurity being the most common complication (*n* = 7, 43.8%). Four (25%) children were born before 33 WG. We recorded three (18.8%) neonatal deaths resulting from one twin and one singleton pregnancy. Twin pregnancy was delivered in 26 WG due to premature rupture of membranes, and babies also had morphological abnormalities: hypotrophy, abnormalities of lower and upper limbs, and calcification around the stomach (probably due to a genetic disorder which was not specified). They died one month after delivery. The third neonatal death occurred in a patient with placental hematoma and ablation. Cesarean section was performed at 27 WG, and the child died due to complications of extreme prematurity. Two (12.5%) other children were diagnosed with congenital malformations—one child presented with patent ductus arteriosus, and the other was born with a diaphragmatic hernia (Table 2).

### 4.3. Obstetrical Status

Obstetrical complications were present in 6 (40%) out of 15 completed pregnancies, among which 1 was multiparous. Two (13.3%) pregnancies were complicated with abruptio placentae. In another two (13.3%), premature rupture of membranes occurred. Antepartum hemorrhage complicated two (13.3%) pregnancies and both patients were followed up using prophylactic anticoagulation. Preeclampsia was not observed in our study group. All the above-mentioned complications resulted in premature delivery before 37 WG (Table 3).

### 4.4. Maternal Cardiovascular Complications

There was neither maternal death nor heart failure decompensation during pregnancy. However, episodes of new-onset arrhythmia occurred. In two (13.3%) women, atrial arrhythmia developed. One patient presented with atrial flutter in the second trimester and required anticoagulation and beta-blocker therapy. The other who had five pregnancies presented with atrial fibrillation during the second pregnancy (this woman suffered three miscarriages and died five years after her last pregnancy because of heart failure). One (6.7%) patient in the second trimester developed ventricular arrhythmia (premature ventricular contractions), which resolved after delivery. She did not require medications. New onset of hypoxemia was observed in one patient with classic (atrio-pulmonary) Fontan palliation (SO_2_—89%) when atrial fibrillation occurred and complicated her second pregnancy. As late maternal complications, we observed Takotsubo syndrome in relation to abruptio placentae and premature delivery of a baby who died subsequently. Four years after delivery, one (6.7%) patient suffered from a pulmonary embolism (Table 4).

### 4.5. Anticoagulation Regimen during Pregnancy

There were three different types of anticoagulation used in our population. Twelve (46.2%) pregnancies were followed without anticoagulation therapy. One of them was treated with aspirin 75mg/d. Prophylactic anticoagulation was administered during six (23%) pregnancies due to abnormal (accelerated in comparison to previous examinations) flow in the TCPC tunnel as well as patent fenestration with a right to left shunt. Therapeutic anticoagulation was used in eight (30.8%) pregnancies due to atrial arrhythmia or thromboembolic complications, diagnosed by abnormal perfusion in pulmonary scintigraphy examination or presence of thrombus in the echocardiographic examination (Table 3).

## 5. Discussion

Our analysis confirmed that pregnancy in Fontan survivors is possible and may also be successful [1,5,7,13]. However, a woman considering pregnancy must be aware of potential risks and complications for herself and the baby. The earliest is a miscarriage, which happened in the analyzed population in 11 (42%) pregnancies, all but one in the first trimester. These observations were the same as described by Bonner et al. [14] and were similar to the incidence of 27–69% published in previous reports [1,5,7,13,15,16]. These numbers are much higher than the spontaneous miscarriage rates of 10–15% observed in the general European population [17]. The reason for this fatal condition is not completely clear, but it may result from abdominal venous congestion, increased venous pressure, and inability to increase cardiac output [1,5,6,18].

As we already know from published data, pregnancies that prevail into the second trimester should be eventually completed [13,14,16,19]. Therefore, it is reasonable to assume that well-functioning Fontan patients are more likely to achieve pregnancy. This was confirmed by Arif et al. based on an analysis of 55 pregnancies in women after the Fontan procedure [16]. These authors proposed a novel three-step risk stratification model based on patients’ functional class and the presence of complications typical for Fontan circulation. They observed a high (93.3%) miscarriage rate among women with poor cardiovascular status. Our data showed similar results. Eleven patients who suffered miscarriage had worse cardiovascular status than women who completed their pregnancies.

### 5.1. Maternal Cardiovascular Complications

What is worth underlining, fortunately, and consistently with published data, is that we did not observe any maternal deaths in our study [1,5,14,16,20].

Our study’s most frequent Fontan-related complication was atrial and ventricular arrhythmia, which occurred in three (20%) patients and responded well to pharmacotherapy.

In the reported literature, supraventricular arrhythmia is the most common cardiovascular adverse event in these pregnant women [1]. According to the published data, an arrhythmia occurs, similarly to our observations, in 8.1–31.5% of patients [16,20,21]. Women after the Fontan procedure are prone to arrhythmia development not only due to the atrial surgical scars, sinus node disease, or altered and injured atrial tissue organization but also as a result of typical changes in the pregnant state, i.e., hyperdynamic circulation, myocardial stretch, or hormonal changes (progesterone increase) [1,13,16,22,23].

None of our patients suffered from heart failure during pregnancy, unlike the published series, where 9.5–15.7% of pregnancies were complicated by heart failure. [16,20,21]. These observations result from our patients’ good clinical and hemodynamical status. This is consistent with the conception made by Arif et al., which states that successful pregnancy is strictly related to the preconceptional clinical state of women after the Fontan procedure [16]. Heart failure was the second most common cardiovascular complication in their studied population. Hence, hyperdynamic circulation typical for pregnancy and resulting from increased cardiac output and dilation of vascular bed impairs the functioning of fragile Fontan circulation. A single ventricle must create such energy to enable blood flow into pulmonary circulation against decreased preload and gravity [18].

Furthermore, obstetrical complications, such as gestational hypertension, preeclampsia, or anemia, may result in heart failure development.

In the analyzed group, in accordance with other published series, we did not observe any new systemic thromboembolism during pregnancy [16,21]. Considering the presence of both thromboembolic risk in women after the Fontan procedure and the hypercoagulable state of pregnancy, patients at risk of thromboembolic complications (presence of arrhythmia, right to left shunt) received anticoagulation in our study. Cauldwell et al. demonstrated only one (out of 50 pregnant women) venous thromboembolism case in the postpartum period [23]. In the series from Gouton et al. as well as Ropero et al., only isolated cases of antepartum pulmonary embolism and cerebral ischemic events were reported [1,20]. In relation to the presented thromboembolic risk, these results require validation in more extensive studies. Initiation of anticoagulation in the studied group must balance the potential risk of bleeding, which is present even in nonpregnant patients [24].

In our study group, all women were of NYHA ≤ II class, and none of the patients had a mechanical valve or pulmonary hypertension. In this setting, the identification of women at risk of primary cardiovascular events using ZAHARA or CARPREG II scores was not particularly discriminative [11,12]. Additionally, the low incidence of cardiovascular complications in our patients could not prove the strength of these risk scores.

### 5.2. Obstetrical Status

Although European guidelines do not give explicit recommendations regarding the delivery mode, all our patients were delivered by semi-elective cesarean section [7]. Almost in half of the patients (6 out of 15), deliveries were conducted before the 37 WG, which significantly increased the risk of potential peri-delivery complications. Similar decisions were made by Arif et al. [16]. Organization of cardiological and obstetrical care in our unit was an additional factor in favor of cesarean section. In the centers participating in our study, obstetrical departments are located in the other part of the city, rendering hours-long high-specialty cardiological and obstetrical care impossible.

As noted in our analysis and described in previous studies, such obstetrical complications as abruptio placentae, premature rupture of membranes, and antepartum hemorrhage are observed significantly more often in pregnancies with Fontan circulation than in the general population [1,20,21,25]. In our study, we had two (13.3%) cases of each complication. In other series, the rate of premature rupture of membranes amounted to 15.3–28.2% vs. 1.25% found in the general population [23,25]. The main reason for this high incidence of complications is hypoperfusion of the fetoplacental unit and vascular dysfunction resulting from the existing Fontan hemodynamic inability to increase stroke volume together with systemic venous hypertension [5]. Antepartum hemorrhage was described to complicate 5.7–21% of pregnancies of women after the Fontan procedure [20,21,23]. We did not observe postpartum hemorrhage, but others reported that complication in 2.8–42.8% of pregnancies in Fontan survivors [20,21,23]. Vascular malformations, liver disease due to abdominal venous congestion, or thrombotic disorders typical for Fontan circulation could be responsible for this observation [24,26]. Not without meaning when postpartum hemorrhage is considered is a mode of delivery, and its protocol, i.e., common use of oxytocin to avoid uterus atony—a potential cause for hemorrhagic complications [27]. In addition, anticoagulation treatment during pregnancy may play a role [21]; however, not all authors support this finding [15,25]. Although current European guidelines suggest consideration of anticoagulation treatment during pregnancy, patients analyzed in our retrospective study were anticoagulated only when additional indications for anticoagulation, i.e., atrial arrhythmia, right to left shunt, or previous thromboembolic events occurred [7]. As was already mentioned, and what needs to be considered while planning pregnancy, anticoagulation can entail miscarriage in these patients.

### 5.3. Neonatal Complications

In our study group, we observed three (18.8%) neonatal deaths, prematurity amounted to 40% of live births, and the birth weight was smaller than in the general population. These results are comparable to other published reports of neonatal deaths at 2.7–25% [20,23,25]. In the other series, prematurity rates in Fontan patients amounted to 60–82% [14,21,23,28] and were six–eight-fold more frequent than in the general population [25]. Presented neonatal complications arise from the above-mentioned adverse hemodynamics regarding Fontan circulation physiology [18,28]. Another predisposing factor for neonatal outcomes could be desaturation due to fenestration, which was present in 40% of patients [20], although Caudwell et al. do not support this thinking [23]. Certainly, maternal medications also play a role, i.e., beta-blockers or diuretics decreasing placental perfusion [1,28].

### 5.4. Management of Pregnant Patients in Our Center

When a patient reaches adulthood and is transferred to our outpatient clinic for adults with congenital heart defects, we analyze previous medical history and talk about reproductive plans. If a patient is considered as “failing” Fontan according to medical records, we discourage her from pregnancy. At preconception evaluation, we perform the echocardiographic examination, laboratory tests (i.e., morphology, NTproBNP, liver function), and cardiopulmonary exercise test. Once a patient gets pregnant, she is asked to come for a visit. Usually, we invite patients once in every trimester for a clinical visit comprising ECG, oxygen saturation, blood pressure, and echocardiographic examination. If additional problems occur (arrhythmia, heart failure symptoms, obstetrical complications), we schedule further visits or hospitalize the patient. Between 20–22 weeks gestation, we advise obtaining a fetal echocardiographic examination. Our patients are comanaged by obstetricians whom we sensify to careful placental and uterus perfusion assessment and thorough evaluation of fetal growth. Usually, we recommend delivery between 37–39 weeks gestation, but its timing depends on the advancement of fetal growth and the presence of potential obstetrical complications. Within three months after delivery, we ask the patient to schedule a clinical visit.

### 5.5. Study Limitations

Single ventricle anatomy and physiology encompasses a spectrum of various defects that may alter Fontan hemodynamics and outcomes. Therefore, the heterogeneity of the studied population makes any comparisons difficult. In addition, this study is limited by the retrospective nature of data collection. Finally, the small sample size is another limitation of this study.

## 6. Conclusions

Pregnancy in women after the Fontan procedure, although it increases the risk of morbidity, is well tolerated. However, it is burdened by a high risk of miscarriage and multiple obstetrical complications, including losing a child, which a young woman and her family should be counseled about. These women require specialized care provided by both experienced cardiologists and obstetricians.

## Figures and Tables

**Table 1 jcm-12-00783-t001:** Pre-conception baseline maternal characteristic.

Population Characteristics	*n* = 15
Type of congenital heart defect	
Tricuspid atresia	5 (33.3%)
Double inlet left ventricle	4 (26.7%)
Pulmonary atresia	3 (20%)
Double outlet right ventricle	3 (20%)
Type of palliation	
Atrio-pulmonary connection	1 (6.7%)
Total cavo-pulmonary connection	14 (93.3%)
Fenestration	6 (40%)
Functional state	
NYHA I/II/III/IV	3/12/0/0
Hypoxemia (<90%)	1 (6.7%)
Past medical history	
Sick sinus syndrome	4 (26.7%)
Atrial arrhythmia	4 (26.7%)
Ventricular arrhythmia	2 (13.3%)
Permanent pacemaker	2 (13.3%)
Thromboembolic complications	2 (13.3%)
Age at first pregnancy (years) mean (SD)	25.4 (3.5)
Incidence of patient’s visits median (range)	3 (2–6)
Echocardiography	
SV morphology	RV—3 (20%), LV—12 (80%)
SV function normal/impaired	14 (93.3%)/1 (6.7%)
Atrio-ventricular regurgitation mild/moderate/severe	10 (66.7%)/5 (33.3%)/0
Thrombus	1 (6.7%)

LV—left ventricle; NYHA—New York Heart Association scale; RV—right ventricle; SV—single ventricle.

**Table 2 jcm-12-00783-t002:** Fetal/neonatal complications—*n* = 16 (one twin pregnancy).

Birth Weight (g) Mean (SD)	2519 (758)
Termination of pregnancy <20 WG	0
Stillbirth	0
Neonatal death	3 (18.8%)
Small for gestational age	1 (6.25%)
Congenital malformations	2 (12.5%)
Prematurity	7 (43.8%)
33–36 + 6 WG	3
28–32 + 6 WG	1
22–27 + 6 WG	3

WG—week of gestation.

**Table 3 jcm-12-00783-t003:** Maternal cardiovascular and obstetrical complications.

Pt	Age at Fontan (Years)	Age at First P (Years)	CARPREG II	ZAHARA	P	M	L	Anticoag	NYHA	Arrhythmia before 1st Pregnancy	Complications during Pregnancy
1	1	29	5	4	1	0	1	Prophyl	II	AA	None
2	4	24	3	6.25	1	0	1	Prophyl	II	AA	PD, AH
3	5	27	8	4.75	1	1	0	Therap	II	SSS, VA	VB
4	7	26	3	4.75	1	0	1	Prophyl	II	SSS	PD; AP; Takotsubo syndrome
5	14	23	3	4.75	3	2	1	Therap	II	SSS	1- Miscarriage 2- Miscarriage 3- AFl
6	4	24	2	2.5	2	1	1	ASA	II	-	1- Miscarriage 2- None
7	5	17	2	2.5	5	3	2	None -1st Therap -2nd -5th pregnancy	II	-	1- None 2- Miscarriage- AF, hypoxemia 3- Miscarriage 4- Miscarriage 5- PD, AP
8	12	24	5	4.75	2	1	1	None	II	AA, VA	1- Miscarriage 2- None
9	5	22	3	2.5	1	0	1	None	I	AA	VA
10	13	28	2	2.5	1	0	1	None	II	-	None
11	5	28	0	1	1	0	1	None	I	-	None
12	7	28	5	5.5	3	2	1	None -1st -2nd Prophyl -3rd pregnancy	II	SSS	1- Miscarriage 2- Miscarriage 3- PD, AH
13	8	23	2	0	2	1	1	None	I	-	1- PD, PROM 2- Miscarriage
14	15	30	2	1.75	1	0	1	Prophyl	II	-	PD, PROM
15	2	29	0	1	1	0	1	Prophyl	I	-	None

AA—atrial arrhythmia; AF—atrial fibrillation; AFl—atrial flutter; AH—antepartum hemorrhage; Anticoag—anticoagulation; AP—abruptio placentae; ASA—aspirin; L—live birth; CARPREG II—the risk score of primary maternal cardiac event, calculated for the first pregnancy, was 5% with a score of 1, 10% for a score of 2, 15% with a score of 3, 22% for a score of 4, and 41% if the score was greater than 4 points; M—miscarriage; NYHA—New York Heart Association; P—pregnancy; PD—preterm delivery; PROM—premature rupture of membranes; Prophyl—prophylactic anticoagulation; SSS—sick sinus syndrome; SV—single ventricle; Therap—therapeutic anticoagulation; VA—ventricular arrhythmia; VB—vaginal bleeding; ZAHARA—the risk score of maternal cardiovascular complications calculated for first pregnancy is 2.9% with <0.5 points, 7.5% with 0.5–1.5 points, 17.5% with 1.51–2.50 points, 43.1% with 2.51–3.5 points, and 70% with >3.5 points.

**Table 4 jcm-12-00783-t004:** Maternal cardiovascular complications during pregnancy and after delivery.

Type of Complication	*n* = 15	Trimester
Atrial arrhythmia	2 (13.3%)	2nd
Ventricular arrhythmia	1 (6.7%)	2nd
Hypoxemia (SO_2_ < 90%)	1 (6.7%)	2nd
Systemic thromboembolism	0	-
Heart failure	0	-
Takotsubo syndrome	1 (6.7%)	After delivery
Protein-losing enteropathy	0	-
Maternal death	0	-

## Data Availability

The data presented in this study are available on request from the corresponding author.

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
