# Peer review of "Pregnancy Outcomes in Women after the Fontan Procedure"

_jcm, 2023, doi:10.3390/jcm12030783_

Round 1

Reviewer 1 Report

In the present study, Bartczak-Rutkowska et al. report their experience with pregnancy in patients with a Fontan circulation. Overall, the findings are consistent with the previously reported results. While maternal mortality is very low, with no such events in the current study, morbidity (fetal, maternal, obstetrical) is significant.

Introduction:

Page 2, line 41: Instead of “high systemic vascular pressures” I suggest saying “high systemic venous pressures”

Materials and Methods:

If the authors have data on preeclampsia prevalence, I suggest they add this information.

Statistics:

In the result section, the authors sometimes report mean with standard deviation and sometimes median with range or interquartile range (not clear which one). This should be clarified in the statistical section.

Results:

Are baseline functional characteristics (NYHA classification etc.) based on pre-conception data? Please clarify.

Do the authors have data on invasive Fontan hemodynamics, such as Fontan pressures? If yes, I suggest that they add this information.

If the authors have data to calculate the CARPREG II and ZAHARA risk scores, I suggest that they add this information as it might add to the risk stratification of their patient population.

Obstetrical status: What type of anticoagulation regimen were the patients with antepartum hemorrhage on?

Typo: “befre”, page 5, line 138

Table 1:

Incidence of patient’s visits: What does this mean? I assume the number of outpatient visits during pregnancy. Please clarify.

Were the patients on any cardiovascular drugs prior to/during pregnancy?

Age at first pregnancy / Incidence of patient’s visits: Please clarify how the numbers are reported (I assume it is mean/SD and median/range).

Table 2

It is confusing when in the first row “live births” is reported in bold, and thus is suggested to include all patients in the table, but later in the table number of terminated pregnancies and stillbirths is reported. I suggest reformatting the table.

Figure 1

Figure 1 is difficult to read since patient baseline characteristics are reported for the patients with miscarriage (but not for the patients with live births) and obstetrical complications are reported for the patients with live births. If the outcomes for miscarriages and live births are reported in the same figure, I suggest reporting comparable parameters for the different pregnancy outcomes. Otherwise, the authors might consider splitting the figure.

The term “live pregnancies” should be replaced.

Anticoagulation regimen during pregnancy:

In many centers, any form of antithrombotic therapy is considered standard for all patients with a Fontan circulation. In the present study, a high number of patients were on no antithrombotic therapy. It would be interesting to have some information on the study center's approach to antithrombotic therapy in patients with a Fontan circulation.

Why do the authors think that accelerated flow in the TCPC conduit is an indication of prophylactic anticoagulation? My understanding is that the risk for thrombus formation is lower in the case of a high-flow state.

Discussion:

It might be interesting to add a paragraph in which the authors describe their management plan for patients with a Fontan circulation and pregnancy. Do they regularly do pre-pregnancy counseling? Do they routinely perform invasive hemodynamic assessment prior to pregnancy? How frequently do they see patients with an uncomplicated course?

Study limitation

While I agree that a small sample size is frequently a limitation in studies on patients with congenital heart disease, I would avoid calling this “standard”. Registries, population-based studies, and multicenter studies are possible even in patients with rare diseases.

Author Response

A point-by-point response to the reviewer’s comments 

Reviewer 1

Introduction:

Page 2, line 41: Instead of “high systemic vascular pressures” I suggest saying “high systemic venous pressures”

Changed according to reviewer’s suggestion

Materials and Methods:

If the authors have data on preeclampsia prevalence, I suggest they add this information.

We do have information on preeclampsia status in our study group- none patient suffered from preeclampsia. This information was added to the manuscript. 

Statistics:

In the result section, the authors sometimes report mean with standard deviation and sometimes median with range or interquartile range (not clear which one). This should be clarified in the statistical section.

Data for continuous variables were expressed as means with SD or median with range (according to normal distribution). This information was added to the manuscript.

Results:

 Are baseline functional characteristics (NYHA classification etc.) based on pre-conception data? Please clarify.

Yes, baseline characteristics is based on pre-conception data with the exception of hypoxemia (2 patients)- one patient was hypoxic before pregnancy, the other started to be hypoxic during her second pregnancy when AF occurred. It is however clarified in the text. I changed headlines in the Table no 1.

Do the authors have data on invasive Fontan hemodynamics, such as Fontan pressures? If yes, I suggest that they add this information.

We don’t have such data, in our center we don’t perform right heart catheterization (RHC) in stable Fontan patients. We do perform RHC in failing Fontans but none of our pregnant patients was in that group.

If the authors have data to calculate the CARPREG II and ZAHARA risk scores, I suggest that they add this information as it might add to the risk stratification of their patient population.

We calculated CARPREG II and ZAHARA risk scores and located the results into Table 3

Obstetrical status: What type of anticoagulation regimen were the patients with antepartum hemorrhage on?

 Antepartum hemorrhage complicated 2 (13.3%) pregnancies, both patients were followed up using prophylactic anticoagulation.

Typo: “befre”, page 5, line 138

Corrected

Table 1:

Incidence of patient’s visits: What does this mean? I assume the number of outpatient visits during pregnancy. Please clarify.

Yes, this means incidence of outpatient visits during pregnancy. Clarified in the table.

Were the patients on any cardiovascular drugs prior to/during pregnancy?

 Besides anticoagulation, three patients were on beta-blockers due to supraventricular arrhythmia prior to pregnancy. Two other patients developed atrial arrhythmia during pregnancy and responded well to beta-blocker therapy.

Age at first pregnancy / Incidence of patient’s visits: Please clarify how the numbers are reported (I assume it is mean/SD and median/range).

Clarified in the Table 1- age at first pregnancy mean/SD, incidence of patient’s visits- median/range

Table 2

It is confusing when in the first row “live births” is reported in bold, and thus is suggested to include all patients in the table, but later in the table number of terminated pregnancies and stillbirths is reported. I suggest reformatting the table.

 The table was reformatted

Figure 1

Figure 1 is difficult to read since patient baseline characteristics are reported for the patients with miscarriage (but not for the patients with live births) and obstetrical complications are reported for the patients with live births. If the outcomes for miscarriages and live births are reported in the same figure, I suggest reporting comparable parameters for the different pregnancy outcomes. Otherwise, the authors might consider splitting the figure.

The term “live pregnancies” should be replaced.

 We changed Figure 1 into Table 3

Anticoagulation regimen during pregnancy:

In many centers, any form of antithrombotic therapy is considered standard for all patients with a Fontan circulation. In the present study, a high number of patients were on no antithrombotic therapy. It would be interesting to have some information on the study center's approach to antithrombotic therapy in patients with a Fontan circulation.

We don’t give anticoagulation to every patient after Fontan procedure. There are no guidelines= no consensus among experts to do so. In our center we start anticoagulation when patient is at risk of thromboembolic complications- ie. with atrial arrhythmia (AF, AFl), after stroke, with impairment in scyntygraphy perfusion of the lungs, with evidence of thrombus in the heart/Fontan tunnel.

We mention it in the manuscript- paragraph anticoagulation regimen during pregnancy- lines 194-202 and Discussion- lines 251-262, 290-295.

Why do the authors think that accelerated flow in the TCPC conduit is an indication of prophylactic anticoagulation? My understanding is that the risk for thrombus formation is lower in the case of a high-flow state.

The flow in the TCPC conduit should be slow, the blood flows into pulmonary circulation, where low pressures are present. When we observe accelerated flow in the TCPC tunnel during echocardiographic examination, this is an indication for narrowing of it- especially if we have comparison to previous examinations, as is stated in the manuscript and usually this narrowing is caused by thrombus formation. Normally, when patient is not pregnant, in such case we move to right heart catheterisation, in the case of pregnant patient we can not perform right heart catheterisation, we start prophylactic anticoagulation.

This assessment of abnormal flow in the TCPC was also mentioned in Gouton M, Nizard J, Patel M, Sassolas F, Jimenez M, Radojevic J, Mathiron A, Amedro P, Barre E, Labombarda F, Vaksmann G, Chantepie A, Le Gloan L, Ladouceur M. Maternal and fetal outcomes of pregnancy with Fontan circulation: a multicentric observational study. Int J Cardiol. 2015, 187, 84–89.

Discussion:

It might be interesting to add a paragraph in which the authors describe their management plan for patients with a Fontan circulation and pregnancy. Do they regularly do pre-pregnancy counseling? Do they routinely perform invasive hemodynamic assessment prior to pregnancy? How frequently do they see patients with an uncomplicated course?

At the end of discussion we added a paragraph describing our management plan for pregnant women after Fontan procedure.

Study limitation

While I agree that a small sample size is frequently a limitation in studies on patients with congenital heart disease, I would avoid calling this “standard”. Registries, population-based studies, and multicenter studies are possible even in patients with rare diseases.

I deleted that it is a standard. As an explanation I want to write, that in comparison to coronary artery disease, where studies encompass thousands of patients, groups with congenital heart defects will never be that big. Very often scientists unfamiliar with the topic of congenital heart disease reproach studies in that population due to small study size. That’s why I wanted to underline that these studies are not that big.

Thank you for your review!

Reviewer 2 Report

The authors present a descriptive study of pregnancy outcomes in patients with Fontan circulation. This is an important clinical concern and having data about pregnancy outcomes is helpful for use in counseling of this patient group. 

Abstract - You state that pregnancy is well tolerated in women with Fontan circulation. It would be helpful to mention here, as you do in the manuscript, something about the baseline health of the patient - ie, that they are a low risk group or have a good baseline clinical status.

Introduction - 

pg1, line 36 - I am not sure what is meant by "finest cardiosurgical interventions".

The introduction provides a good summary of the background information and reasoning for the study.

Results - 

pg4, line 109-111 - The hypoxemia for the patient with the fenestration seems like a baseline findings, but was the hypoxemia for the other patient present at baseline or did it develop during the pregnancy or between pregnancies? If it developed during the pregnancy it might be better to include under the cardiovascular outcomes. 

pg4, line 128 - was the PDA present just at birth or beyond the immediate postnatal period? May or may not count as a "malformation" depending on the timing of its presence.

The results present a lot of helpful information about the pregnancy outcomes. It would be helpful to have more information about the pregnancy outcomes of the women who had multiple pregnancies, as you present for the woman who had 5 pregnancies. It would help the reader understand if there were some women having multiple bad outcomes, or if they had one bad outcome and then a successful pregnancy, etc. It could be described in the text or since there are only 15 patients it could be a table with each patient and their pregnancy outcome(s) instead of Figure 1.

Discussion - 

pg8, line 262-64 - I do not understand the use of the phrase "not without meaning".

Overall, the discussion looks good.

Author Response

The authors present a descriptive study of pregnancy outcomes in patients with Fontan circulation. This is an important clinical concern and having data about pregnancy outcomes is helpful for use in counseling of this patient group. 

Abstract - You state that pregnancy is well tolerated in women with Fontan circulation. It would be helpful to mention here, as you do in the manuscript, something about the baseline health of the patient - ie, that they are a low risk group or have a good baseline clinical status.

Pregnancy itself poses a hemodynamic burden for distorted physiology of Fontan circulation but according to literature it is usually well tolerated unless patient is a “failing” Fontan. – added as suggested

Introduction - 

pg1, line 36 - I am not sure what is meant by "finest cardiosurgical interventions".

We used this wording to express greatness of Fontan concept/procedure- cardiosurgical interventions that enable people earlier dying in childhood reach adulthood, give birth.. our oldest Fontan patient is now 44 years old and still in good clinical state, without any complications so far.. for the first time, at this moment, we are following two adult patients with HLHS who reached adulthood and one has got plastic bronchitis, but the other is in good shape. This never happened before.. This concept that human being can live without subpulmonary ventricle was outbreaking and in our opinion- as doctors dealing with adults with CHD- is the most complex set of cardiosurgical interventions with such a good effect.

I changed the wording to :

The Fontan procedure is one of the most inventive cardiosurgical concepts.

The introduction provides a good summary of the background information and reasoning for the study.

Results - 

pg4, line 109-111 - The hypoxemia for the patient with the fenestration seems like a baseline findings, but was the hypoxemia for the other patient present at baseline or did it develop during the pregnancy or between pregnancies? If it developed during the pregnancy it might be better to include under the cardiovascular outcomes. 

In the first patient hypoxemia was resultant to fenestration and was present at baseline. The other patient with classic Fontan procedure developed hypoxemia during second pregnancy due to atrial fibrillation. I included this to cardiovascular outcomes.

pg4, line 128 - was the PDA present just at birth or beyond the immediate postnatal period? May or may not count as a "malformation" depending on the timing of its presence.

The child is now 1 year old and PDA is still present.

The results present a lot of helpful information about the pregnancy outcomes. It would be helpful to have more information about the pregnancy outcomes of the women who had multiple pregnancies, as you present for the woman who had 5 pregnancies. It would help the reader understand if there were some women having multiple bad outcomes, or if they had one bad outcome and then a successful pregnancy, etc. It could be described in the text or since there are only 15 patients it could be a table with each patient and their pregnancy outcome(s) instead of Figure 1.

As suggested we changed Figure 1 into Table, describing pregnancy outcomes in each patient.

Discussion - 

pg8, line 262-64 - I do not understand the use of the phrase "not without meaning".

In our center we deliver Fontan pregnancies by semi-elective caesarean section and usually we use oxitoticin afterwards. In this manner we hinder postpartum hemorrhage and in this management we see a cause of no postpartum hemorrhage in our study group.

I changed it to:

Not without meaning when postpartum hemorrhage is considered is a mode of delivery and its protocol, i.e., common use of oxytocin to avoid uterus atony as a potential cause for hemorrhagic complications [25].

Overall, the discussion looks good.

Thank you for the review!